

# Ligand determinants of fatty acid activation of the pronociceptive ion channel TRPA1

William John Redmond[1], Liuqiong Gu[2], Maxime Camo[1], Peter McIntyre[2,3] and Mark Connor[1]

[1] Australian School of Advanced Medicine, Macquarie University, NSW, Australia
[2] Department of Pharmacology, University of Melbourne, Parkville, Victoria, Australia
[3] Health Innovations Research Institute and School of Medical Sciences, RMIT University, Melbourne, Victoria, Australia

## ABSTRACT

**Background and purpose.** Arachidonic acid (AA) and its derivatives are important modulators of cellular signalling. The transient receptor potential cation channel subfamily A, member 1 (TRPA1) is a cation channel with important functions in mediating cellular responses to noxious stimuli and inflammation. There is limited information about the interactions between AA itself and TRPA1, so we investigated the effects of AA and key ethanolamide and amino acid/neurotransmitter derivatives of AA on hTRPA1.

**Experimental approach.** HEK 293 cells expressing hTRPA1 were studied by measuring changes in intracellular calcium ($[Ca]_i$) with a fluorescent dye and by standard whole cell patch clamp recordings.

**Key results.** AA (30 $\mu$M) increased fluorescence in hTRPA1 expressing cells by 370% (notional $EC_{50}$ 13 $\mu$M). The covalent TRPA1 agonist cinnamaldehyde (300 $\mu$M) increased fluorescence by 430% ($EC_{50}$, 11 $\mu$M). Anandamide (230%) and $N$-arachidonoyl tyrosine (170%) substantially activated hTRPA1 at 30 $\mu$M, however, $N$-arachidonoyl conjugates of glycine and taurine were less effective while $N$-acyl conjugates of 5-HT did not affect hTRPA1. Changing the acyl chain length or the number and position of double bonds reduced fatty acid efficacy at hTRPA1. Mutant hTRPA1 (Cys621, Cys641 and Cys665 changed to Ser) could be activated by AA (100 $\mu$M, 40% of wild type) but not by cinnamaldehyde (300 $\mu$M).

**Conclusions and implications.** AA is a more potent activator of TRPA1 than its ethanolamide or amino acid/neurotransmitter derivatives and acts via a mechanism distinct from that of cinnamaldehyde, further underscoring the likelyhood of multiple pharmacologically exploitable sites on hTRPA1.

Corresponding author
William John Redmond,
william.redmond@mq.edu.au

## INTRODUCTION

The transient receptor potential ankyrin 1 channel (TRPA1; *Alexander, Mathie & Peters, 2011*, *Story et al., 2003*) is expressed on primary afferent nociceptors where it detects potentially damaging environmental stimuli such as noxious cold, changes in pH, noxious chemicals and endogenous products of inflammation. Although there is emerging evidence for physiological roles of TRPA1 in cells intrinsic to brain and spinal cord (*Shigetomi et al., 2011*; *Cho et al., 2012*) and TRPA1 is also expressed in the hair cells of the ear (*Corey et al., 2004*), most effects of TRPA1 ligands have been linked with the expression of the channel in the peripheral sensory neurons of the dorsal root, trigeminal and nodose ganglia (*Nagata et al., 2005*; *Story et al., 2003*).

Although a complete description of how TRPA1 is activated by such a wide variety of modulators is yet to be realized, electrophilic agonists such as cinnamaldehyde (CA) and allyl isothiocyanate (AITC) activate TRPA1 via reversible or irreversible covalent modification of cysteine residues located within the intracellular N-terminal domain (*Hinman et al., 2006*; *Macpherson et al., 2007*). The mechanism(s) underlying the activation of TRPA1 by unreactive compounds such as menthol (*Karashima et al., 2007*), 5-nitro-2-(3-phenylpropylamino)benzoic acid (NPPB) (*Liu et al., 2010*) and $\Delta^9$-tetrahydrocannabinol (*Jordt et al., 2004*) are less well described, although in some cases residues in transmembrane domains appear to be important for channel activation by these ligands (e.g. menthol, *Xiao et al., 2008*). Intriguingly, mutation of cysteine residues which abolish TRPA1 activation by electrophiles appears to also reduce the effectiveness of TRPA1 activation by most unreactive compounds, implying an important general role for these cysteine residues in channel function. For example, a mutagenesis study by (*Liu et al., 2010*) showed that reactive and non-reactive compounds such as NPPB saw their peak $[Ca_i]$ response reduced for single cysteine mutations to a serine. Xiao described that these residues are important for the activation of the channel mediated by menthol, another non-reactive compound (*Xiao et al., 2008*). A requirement for formation of disulphide bonds between cysteine residues during channel activation, including activation by non-reactive compounds, might explain why there is lessened activity in the 3x Cys mutants (*Wang et al., 2012*).

TRPA1 is activated by arachidonic acid-derived molecules, including highly reactive isoprostanes, prostaglandins (*Taylor-Clark et al., 2008*), hepoxilins (*Gregus et al., 2012*), epoxyeicosatreinoic acids (*Sisignano et al., 2012*) and the endocannabinoid anandamide (*De Petrocellis & Di Marzo, 2009*). Arachidonic acid (AA) itself has also been reported to activate TRPA1 (*Bandell et al., 2004*; *Motter & Ahern, 2012*). In this study we have examined the activation of recombinant human TRPA1 (hTRPA1) by arachidonic acid and other long chain fatty acids as well as by *N*-arachidonoyl neurotransmitter/amino acid conjugates (NAAN), a large family of endogenous modulators of ion channels and G protein coupled receptors (*Connor, Vaughan & Vandenberg, 2010*). We find that AA itself is the most effective activator of hTRPA1 among these compounds, and modest changes in its structure dramatically alter TRPA1 activity. Mutations in the intracellular Cys

**Peer**J

residues that essentially abolish the activity of CA also reduce the effects of AA, suggesting some overlap in the mechanisms through which diverse agonists activate the channel.

## METHODS

### Cell culture

Flp-In TRex HEK 293 (Life Technologies, Mulgrave, Victoria, Australia) were stably transfected with wild type or mutant hTRPA1 or wild type mouse TRPA1 (Genscript, Piscataway, NJ, USA) and cultivated in Dulbecco's modified Eagle's Medium supplemented with 10% fetal bovine serum, 100 U penicillin and 100 $\mu$g streptomycin ml$^{-1}$, hygromycin B 25 $\mu$g ml$^{-1}$ and blasticidin S 5 $\mu$g ml$^{-1}$. Cells were incubated in 5% $CO_2$ at 37°C in a humidified atmosphere. Cells were grown in flasks with a surface area of 75 mm$^2$, once at optimum confluence (approximately 90%), cells were trypsinized and transferred into clear-bottomed poly-D-lysine coated 96 well plates (Corning, Castle Hill, NSW, Australia) in L15 medium supplemented with 1% fetal bovine serum, hygromycin B, and the antibiotics outlined above. The cells were plated in a volume of 100 $\mu$L and were incubated in humidified room air at 37°C overnight. Expression of the TRPA1 receptor or mutants was induced 5–8 h prior to experimentation by addition of with tetracycline, 1 $\mu$g ml$^{-1}$ to each well.

### Calcium assay

Intracellular calcium $[Ca]_i$ was measured with the calcium 5 kit from Molecular Devices (Sunnyvale, CA, USA) using a FLEX Station 3 Microplate Reader (Molecular Devices, Sunnyvale, CA, USA). 100 $\mu$l of dye dissolved in HEPES- buffered saline (HBS) containing (in mM): NaCl 140, KCl 5.33, $CaCl_2$ 1.3, $MgCl_2$ 0.5, HEPES 22, $Na_2HPO_4$ 0.338, $NaHCO_3$ 4.17, $KH_2PO_4$ 0.44, $MgSO_4$ 0.4, glucose 10 (pH to 7.3, osmolarity $= 330 \pm 5$ mosmol) was loaded into each well of the plate for 1 h prior to testing in the Flexstation at 37°C. Fluorescence was measured every 2 seconds ($\lambda_{excitation} = 485$ nm, $\lambda_{emission} = 525$ nm) for the duration of the experiment. Drugs were added after at least 2 min of baseline recording. In experiments where one drug addition was made, 50 $\mu$L of drug dissolved in HBS was added, for two drug additions, 25 $\mu$L was added each time.

### Electrophysiology

TRPA1 channel currents in HEK293 cells were recorded in the whole-cell configuration of the patch-clamp method (*Hamill et al., 1981*) at room temperature. Dishes were perfused with HEPES buffered saline (HBS) containing (in mM): 140 NaCl, 2.5 KCl, 2.5 $CaCl_2$, 1 $MgCl_2$, 10 HEPES, 10 Glucose (pH to 7.3, osmolarity $= 330 \pm 5$ mosmol). Recordings were made with fire-polished borosilicate glass pipettes with resistance ranging from 2–3 M . The internal solution contained (in mM): 130 CsCl, 10 HEPES, 2 $CaCl_2$, 10 EGTA, 5 MgATP (pH to 7.3, osmolarity $= 285 \pm 5$ mosmol). Recordings were made with a HEKA EPC 10 amplifier with Patchmaster acquisition software (HEKA Elektronik, Germany). Data was sampled at 10 kHz, filtered at 3 kHz, and recorded on hard disk for later analysis. Series resistance ranged from 3 to 10 M , and was compensated by at least 80% in all experiments. Leak subtraction was not used. Cells were

exposed to drugs via flow pipes positioned approximately 200 $\mu$m from the cell, drugs were dissolved in HBS immediately before application. All solutions had final ethanol concentration of 0.05%–0.1% v/v.

### Data analysis

The response to agonists was expressed as a percentage change over the baseline averaged for the 30 seconds immediately prior to drug addition. Changes produced by parallel solvent blanks were subtracted before normalization, these changes were never more than 10% of baseline. Concentration-effect data were fit to a four-parameter logistic Hill equation to derive the $EC_{50}$ values and Hill slope (GraphPad Prism, San Diego, CA). Where solubility precluded determining full concentration response curves, the curve maxima were constrained to the maximum increase in $[Ca]_i$ produced by a high concentration of cinnamaldehyde in the same experiment. In these cases drug potency was reported as a notional $EC_{50}$. Comparisons between human and mouse TRPA1 were made after normalising responses to those produced by a maximally effective concentration of CA (300 $\mu$M) included in each experiment. Results are expressed as mean $\pm$ s.e.m. of at least 4–5 independent determinations.

### Drugs and reagents

All drugs were made up in ethanol and diluted in HBS to give a final concentration of ethanol of 0.05–0.1%. Because of limits to the solubility of fatty acids and their derivatives, the maximum concentration used was either 30 $\mu$M or 100 $\mu$M, as noted. Arachidonic acid and its derivatives were purchased from Cayman Chemical (Ann Arbor, MI, USA) and NAAN were purchased from Biomol (Plymouth Meeting, PA, USA) or Cayman Chemicals. NPPB was purchased from Tocris Bioscience (Bristol, UK), ruthenium red from Enzo Lifesciences (Farmingdale, NY, USA), HC 030031 and ionomycin were from Ascent Scientific (Avonmouth, UK). Cinnamaldehyde was purchased from Sigma-Aldrich (Castle Hill, NSW, Australia). All tissue culture reagents were from Sigma-Aldrich, Life Technologies (Mulgrave, Victoria, Australia) or Invivogen, (San Diego, CA, USA).

To independently confirm the activity of adrenic acid and $\omega$3-arachidonic acid, we compared their effects on $Ca_V3.1$ calcium channels with those of $\omega$6-arachidonic acid. Recordings from $Ca_V3.1$ channels were made as described in (*Gilmore et al., 2012*). Cells were stepped repetitively from $-86$ mV to $-26$ mV for 20 ms every 10 s. At a concentration of 10 $\mu$M ($n = 3$ each), adrenic acid ($64 \pm 8\%$), $\omega$3-arachidonic acid ($43 \pm 5\%$) and $\omega$6-arachidonic acid ($81 \pm 4\%$) all inhibited $Ca_V3.1$ channels to a degree consistent with previous reports of fatty acid activity on this channel (*Chemin, Nargeot & Lory, 2007*).

## RESULTS

Arachidonic acid has previously been reported to activate mouse and rat TRPA1 expressed in Chinese hamster ovary and HEK 293 cells respectively (*Bandell et al., 2004*; *Motter & Ahern, 2012*). Addition of AA to HEK 293 cells expressing hTRPA1 produced

rapid and sustained elevations of $[Ca]_i$. The increase was concentration-dependent, in our initial series of experiments AA (30 $\mu$M) increased cellular fluorescence by 369 $\pm$ 38% over baseline with a notional $EC_{50}$ of 13 $\pm$ 4 $\mu$M ($n = 5$). Concentration-response curves were fitted based on the assumption that AA had a similar effect to the highest concentration of CA we used in our experiments. Cinnamaldehyde (*Bandell et al., 2004*) activated hTRPA1 with an $EC_{50}$ of 11 $\pm$ 2 $\mu$M, producing a maximum change of fluorescence of 431 $\pm$ 29% at 300 $\mu$M ($n = 8$) (Fig. 1). We were reluctant to use higher concentrations of CA because of the possibility of unspecific effects on the cells. Since these studies were completed, it has been reported that at concentration higher than 300 $\mu$M, CA has complex effects on TRPA1 reflecting both activation and inhibition of the channel (*Alpizar et al., 2013*). It is not possible to study this complexity using our experimental design. In our experiments, CA provides a constant reference response between experiments. Addition of the non-selective antagonist of TRPA1, ruthenium red (10 $\mu$M), largely blocked the increase of $[Ca]_i$ caused by AA (30 $\mu$M) and CA (300 $\mu$M) (Fig. 2). The specific antagonist of TRPA1, HC-030031 (30 $\mu$M) abolished the responses to 10 $\mu$M AA and 30 $\mu$M CA (Fig. 2). Cells that were not incubated with tetracycline 4–8 h prior to experimentation showed highly attenuated responses (Fig. 2). In order to test whether saturation of dye responses occurred during experiments, the effects of ionomycin (3 $\mu$M), an ionophore which elevates $[Ca]_i$, were determined. The responses to the highest concentrations of CA (300 $\mu$M) and AA (100 $\mu$M) tested were on average 63 $\pm$ 7% and 55 $\pm$ 6% of the response to 3 $\mu$M ionomycin, respectively. This indicates that the maximal TRPA1-mediated signal in our cells does not saturate the reporter dye and that we are working within the dynamic range of our experimental system.

To confirm that AA and CA were activating a membrane conductance, whole cell voltage clamp recordings were made from hTRPA1 HEK 293 cells induced overnight with a low concentration of tetracycline (1 $\mu$g mL$^{-1}$). Whole currents were elicited by repeatedly ramping the membrane potential of the cells from $-80$ mV to $+80$ mV over 500 ms. The holding potential was 0 mV. AA (10 $\mu$M) produced a rapid increase in membrane current measured at $+80$ mV (from a baseline of 280 $\pm$ 10 pA to a peak of 3.6 $\pm$ 1.0 nA, $n = 6$, Fig. 3) that was blocked by co-incubation of the cells with ruthenium red (RR, 10 $\mu$M; control 340 $\pm$ 9 pA; in AA and RR 247 $\pm$ 6 pA, $n = 6$). Superfusion of the cells with CA (100 $\mu$M) produced a similar current (baseline 306 $\pm$ 8 pA, peak 4.6 $\pm$ 1.5 nA, $n = 5$, Fig. 3).

Arachidonic acid can be metabolized to a number of molecules that activate TRPA1. To address the possibility that AA metabolites were mediating the observed effects, we preincubated cells with inhibitors of fatty acid amide hydrolase (FAAH), lipoxygenases and cyclooxygenases. $N$-arachidonoyl serotonin (NA-5HT, FAAH, *Maione et al., 2007*), caffeic acid, (lipoxygenases, *Koshihara et al., 1983*) and aspirin (cyclooxygenase, *Vane, 1971*) were used at a concentration of 10 $\mu$M and preincubated with the cells for 10 min before an addition of 10 $\mu$M arachidonic acid. The effect of AA was not altered by application of these enzyme inhibitors, ($P > 0.3$ for each; Fig. 4), leading us to believe

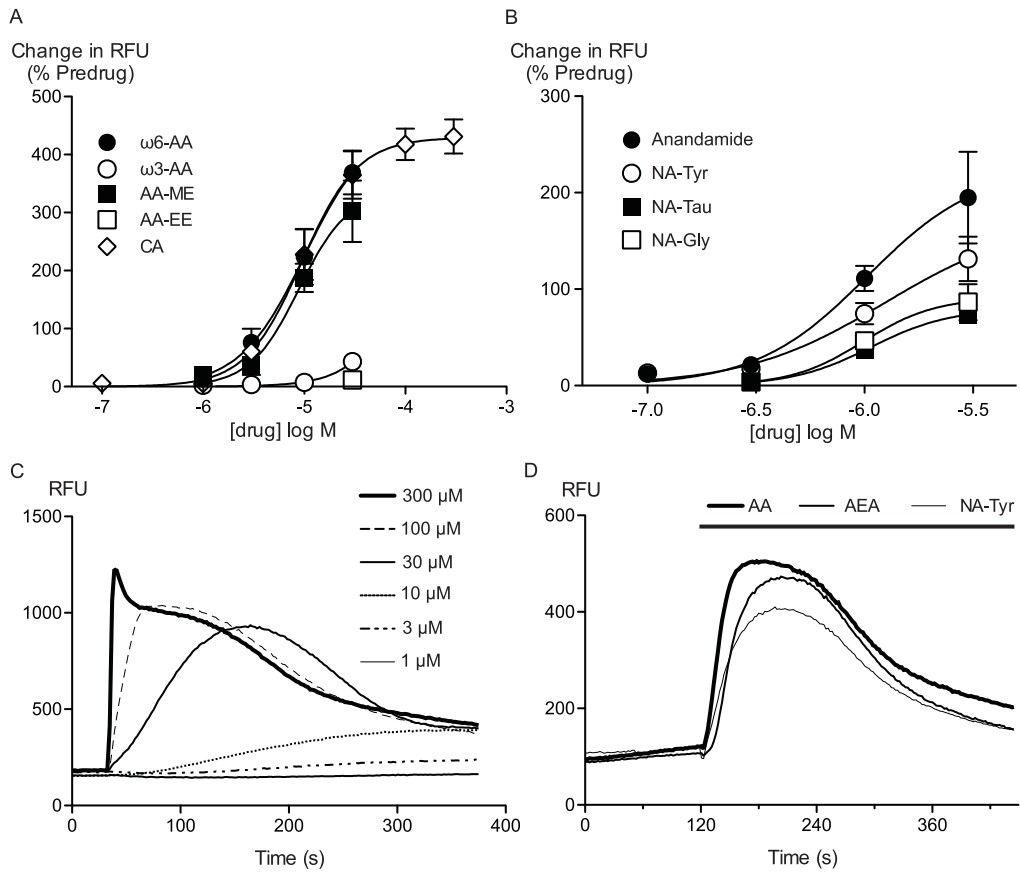

**Figure 1 Arachidonic acid and related molecules elevate calcium in HEK 293 cells expressing hTRPA1.** Changes in intracellular calcium concentration were determined as outlined in the Methods. (A) Concentration response curves for ω6-arachidonic acid (ω6-AA), ω3-arachidonic acid (ω3-AA), arachidonic acid methyl-ester (AA-ME), arachidonic acid ethyl-ester (AA-EE) and cinnamaldehyde (CA) at hTRPA1. Each data point represents the mean ± s.e.m. of 4–5 determinations in triplicate. The curves for AA and CA essentially overlap. (B) Concentration response curves for anandamide, *N*-arachidonoyl tyrosine (NA-Tyr), *N*-arachidonoyl taurine (NA-Tau) and *N*-arachidonoyl glycine (NA-Gly) at hTRPA1. Each data point represents the mean ± s.e.m. of 4–5 determinations in triplicate. (C) Representative traces of change in fluorescence produced by concentrations of CA between 1 $\mu$M and 300 $\mu$M, expressed as raw fluorescence units. CA was applied for the duration of the bar. (D) Representative traces of change in fluorescence produced by 30 $\mu$M anandamide, arachidonic acid and NA-Tyr, expressed as raw fluorescence units. Drugs were applied for the duration of the bar.

that the activation of TRPA1 by AA was direct, and not due to its modification via any of its main metabolic pathways.

We next examined the structural features of arachidonic acid (20:4 ω6) relevant to TRPA1 activation. The relative insolubility of fatty acids meant that determining the maximal possible activation of TRPA1 for most compounds was not possible, and so we chose a fixed concentration of 30 $\mu$M to make comparisons with. Increasing or decreasing the degree of saturation on the fatty acid chain substantially or changing the

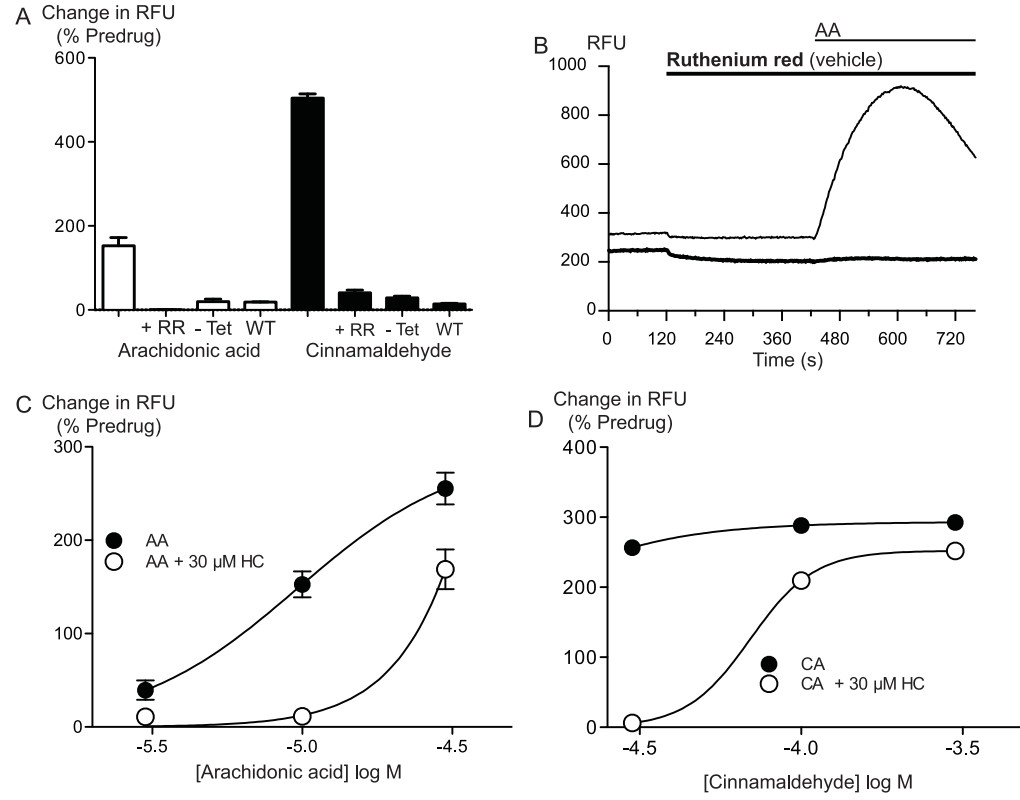

**Figure 2 Arachidonic acid activates hTRPA1.** Changes in intracellular calcium concentration ($[Ca]_i$) was determined as described in the Methods. (A) Elevations of $[Ca]_i$ produced by arachidonic acid (AA, 10 $\mu$M) were absent in Flp-In TRex HEK 293 expressing hTRPA1 not induced with tetracycline, and in untransfected Flp-In TRex HEK 293 cells. The effects of AA (10 $\mu$M) and cinnamaldehyde (CA, 30 $\mu$M) were also strongly reduced by ruthenium red (10 $\mu$M) (representative trace of AA in the presence of RR (B)). HC-030031 (30 $\mu$M), a specific inhibitor of TRPA1, inhibited the elevations of $[Ca]_i$ produced by AA (C) and CA (D) in an apparently competitive manner. Each point represents the mean $\pm$ s.e.m of at least 4 determinations. Error bars within the point for (C).

acyl chain length reduced the capacity of the ligand to activate hTRPA1. Docosahexaenoic acid (DHA 22:6 $\omega$3) and linoleic acid (18:2 $\omega$6) produced modest elevations of $[Ca]_i$ in hTRPA1-expressing HEK 293 cells when applied at 30 $\mu$M (Table 1). Adrenic acid (22:4 $\omega$6), oleic acid (18:1 cis-$\omega$9) and elaidic acid (18:1 trans-$\omega$9) produced changes in $[Ca]_i$ of less than 20% at 30 $\mu$M. Arachidonic acid methyl ester (30 $\mu$M) produced similar elevations of $[Ca]_i$ to AA (30 $\mu$M) (Table 1), however, arachidonic acid ethyl ester (30 $\mu$M) was essentially devoid of agonist activity at hTRPA1. Interestingly, $\omega$6-arachidonic acid had a greater agonist activity at hTRPA1 than $\omega$3-arachidonic acid (Table 1).

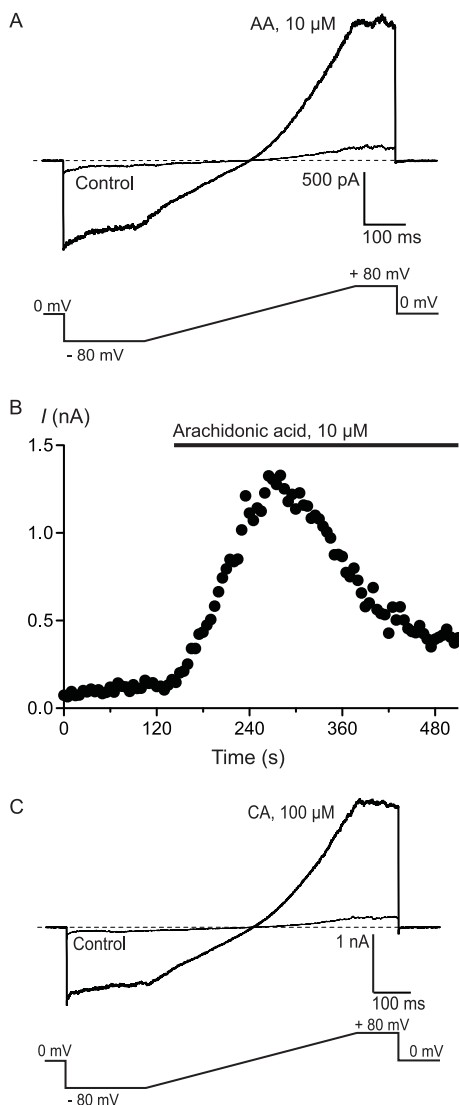

**Figure 3 Arachidonic acid-induced currents in HEK 293 cells expressing hTRPA1.** Whole voltage clamp recordings of membrane currents in HEK 293 cells expressing hTRPA1 were made as outlined in the Methods. (A) Current traces from a hTRPA1-expressing HEK 293 cell in control conditions (thin line) and in the presence of 10 $\mu$M arachidonic acid (AA). Cells were subject to the voltage protocol illustrated beneath the traces. Zero current is designated by the dotted line. (B) A plot of the amplitude of the cell current measured at +80 mV for the same cell, AA was added for the duration of the bar. Typical of 6 similar cells. (C) Current traces from a hTRPA1-expressing HEK 293 cell in control conditions (thin line) and in the presence of 100 $\mu$M cinnamaldehyde (CA). Cells were subject to the voltage protocol illustrated beneath the traces. Zero current is designated by the dotted line. Typical of 5 similar cells.

The first characterization of AA activation of TRPA1 was performed largely with mTRPA1 (*Motter & Ahern, 2012*), and so we compared fatty activation of mTRPA1 with that of hTRPA1 under our experimental conditions. In these experiments the effects of

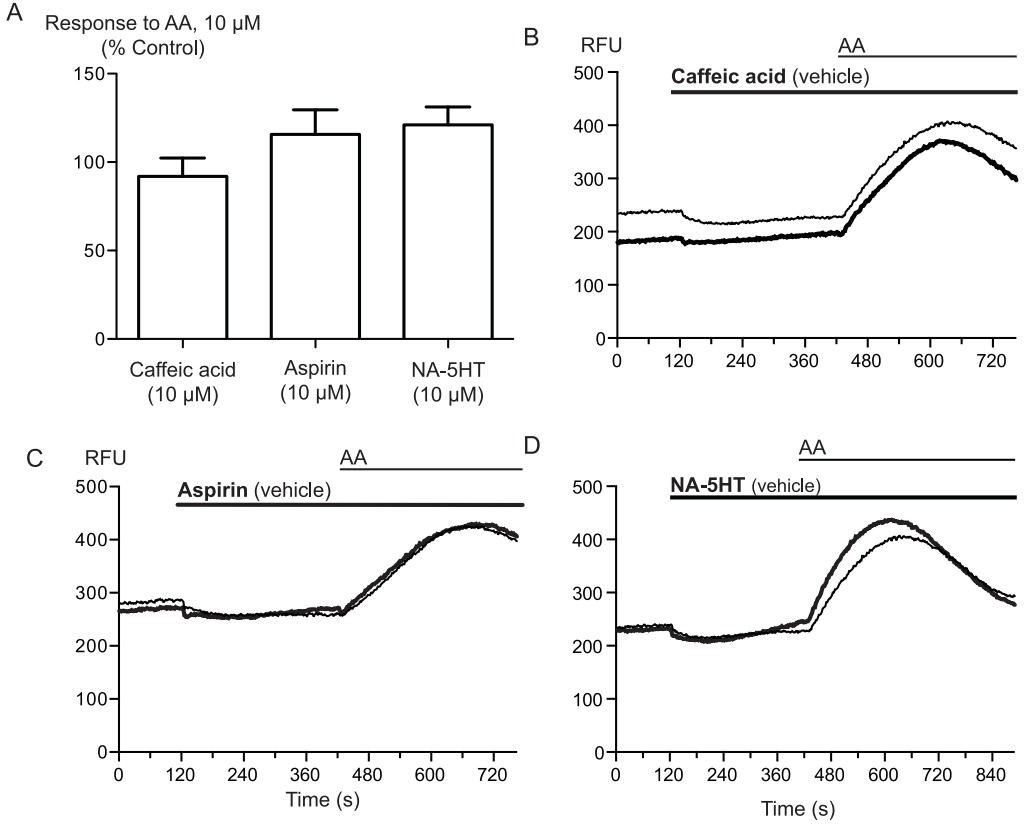

**Figure 4** **Inhibitors of arachidonic acid metabolism do not affect arachidonic acid activation of TRPA1.** (A) Changes in intracellular calcium concentration were determined as described in the Methods. Pre-incubation of cells with inhibitors of lipoxygenase (caffeic acid, 10 $\mu$M), fatty acid amide hydrolase ($N$-arachidonoyl 5-HT, 10 $\mu$M) or cyclooxygenase (aspirin, 10 $\mu$M) did not affect elevations of $[Ca]_i$ produced by 10 $\mu$M arachidonic acid through hTRPA1 ($P > 0.35$ for each). Bar graphs represent the mean s.e.m of at least 8 independent determinations per condition. Representative traces for arachidonic acid by itself or in the presence of caffeic acid (B), aspirin (C) and N-arachidonoyl 5-HT (C) provided. They are respectively inhibitors of the lipoxygenase, cyclooxygenase pathways and an inhibitor of fatty acid amid-hydrolase. Each compound was used at 10 $\mu$M.

fatty acids were normalized to the effect produced by a high (300 $\mu$M) concentration of CA, to control for any differences between the number of channels expressed in the mTRPA1 and hTRPA1 cell lines. AA (100 $\mu$M) produced an increase in $[Ca]_i$ that was $74 \pm 12\%$ of that CA at hTRPA1, and $81 \pm 4\%$ at mTRPA1 ($n = 5$ each, $P > 0.6$). Both DHA (100 $\mu$M, $51 \pm 7\%$ and $32 \pm 6\%$ of CA at hTRPA1 and mTRPA1 respectively, $P = 0.125$) and $\omega$3-AA (100 $\mu$M, $16 \pm 5\%$ and $5 \pm 2\%$ of CA at hTRPA1 and mTRPA1 respectively) activated TRPA1 less than the equivalent concentration of AA.

Amino acid/neurotransmitter conjugates of arachidonic acid are a large group of AA derivatives with an incompletely characterized pharmacology. The prototypic NAAN, $N$-arachidonoyl glycine (NA-Gly) produced modest activation of hTRPA1 at the highest

Table 1 **Activation of hTRPA1 by arachidonic acid and derivatives.** Changes in intracellular calcium concentration were determined as outlined in the Methods. Each compound was applied to HEK 293 cells expressing hTRPA1 at a concentration of 30 $\mu$M. Activation of hTRPA1 by cinnamaldehyde was used as a positive control. The values represent the mean $\pm$ s.e.m. of the percent changes in raw fluorescence units, $n = 4$–$5$ determinations per compound.

| Compound | Change in RFU (% Predrug) |
| --- | --- |
| Cinnamaldehyde (300 $\mu$M) | 426 $\pm$ 28 |
| Arachidonic acid C20:4 $\omega$6 | 369 $\pm$ 38 |
| Arachidonic acid C20:4 $\omega$3 | 43 $\pm$ 10 |
| Arachidonic acid methyl ester | 302 $\pm$ 53 |
| Arachidonic acid ethyl ester | 12 $\pm$ 4 |
| Docosohexaenoic acid C22:6 $\omega$3 | 121 $\pm$ 24 |
| Linoleic acid C18:2 $\omega$6 | 49 $\pm$ 15 |
| Arachidonoyl ethanolamide | 195 $\pm$ 48 |
| Linoleoyl ethanolamide | 12 $\pm$ 5 |
| $N$-arachidonoyl tyrosine | 156 $\pm$ 18 |
| $N$-arachidonoyl taurine | 74 $\pm$ 26 |
| $N$-arachidonoyl glycine | 87 $\pm$ 19 |
| $N$-arachidonoyl dopamine | 32 $\pm$ 6 |
| $N$-oleoyl dopamine | 22 $\pm$ 3 |
| $N$-arachidonoyl 5-HT | 7 $\pm$ 2 |

concentration tested (30 $\mu$M), while the structurally similar endocannabinoid arachidonoyl ethanolamide (anandamide, C20:4 $\omega$6), robustly activated the channel (227 $\pm$ 42% increase in [Ca]$_i$, at 30 $\mu$M, Fig. 1, (*De Petrocellis & Di Marzo, 2009*). By contrast, lineoyl ethanolamide (C18:2 $\omega$6) was essentially inactive at hTRPA1. $N$-arachidonoyl tyrosine (NA-Tyr) also activated TRPA1 to a substantial degree (172 $\pm$ 20% increase in [Ca]$_i$ at 30 $\mu$M) but other NAAN with aromatic head groups, $N$-arachidonoyl dopamine (NA-DA), $N$-oleoyl dopamine (OL-DA) and NA-5HT, were ineffective at 30 $\mu$M (Table 1). $N$-arachidonoyl taurine (NA-Tau) was also a poor activator of TRPA1 (Table 1). OL-DA is also a potent inhibitor of 5-lipoxygenases (*Tseng et al., 1992*), but it failed to inhibit the effects of AA (30 $\mu$M), the elevation of [Ca]$_i$ was 115 $\pm$ 9% by AA alone, and 148 $\pm$ 20% in the presence of 30 $\mu$M OL-DA.

AA is unlikely to activate TRPA1 by covalent modification so we sought to determine whether there were differences between AA and CA activation of TRPA1. We first assessed whether AA and CA could activate the channel in a synergistic manner. Prior administration of subthreshold doses of AA (100 nM, 300 nM or 1 $\mu$M) failed to affect the concentration relationship of subsequently applied CA (1 $\mu$M to 300 $\mu$M, Fig. 5).

We next assessed whether activation of TRPA1 by high concentrations of either AA or CA affected the response to a subsequent addition of the other agonist. Application of either drug produced a robust increase in [Ca]$_i$ which declined over the next 15 to 20 min. Addition of CA (300 $\mu$M) after 30 min of AA (100 $\mu$M) produced a very small increase in [Ca]$_i$, as did another addition of AA (100 $\mu$M) at this point. Similarly, application of AA (100 $\mu$M) following CA (300 $\mu$M) also produced only a small increase in [Ca]$_i$. Thus, each agent produced essentially complete cross-desensitization to the

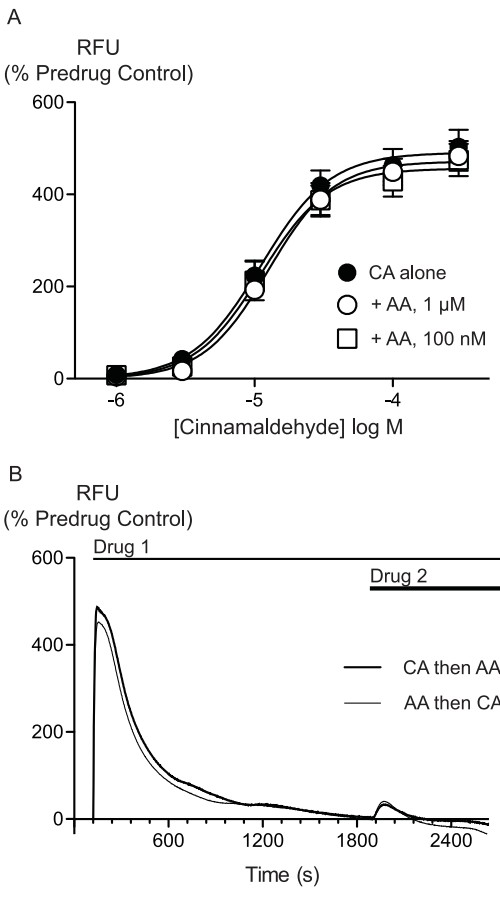

**Figure 5 Lack of interaction between arachidonic acid and cinnamaldehyde in activation of hTRPA1.** Changes in intracellular calcium concentration were determined as outlined in the Methods. (A) Concentration response curves for cinnamaldehyde (CA) in control conditions, and in the presence of arachidonic acid (AA, 100 nM, 1 $\mu$M). Each point represents the mean $\pm$ s.e.m. of 4–5 determinations in triplicate. (B) Example traces showing reciprocal cross-desensitization between CA (300 $\mu$M) and AA (100 $\mu$M). Drugs were applied for the duration of the bars. Traces represent typical data from 4-5 independent replicates per condition. RFU = raw fluorescence units.

other (Fig. 5, Table 2). Ionomycin (30 $\mu$M) administered 30 min after the addition of AA (100 $\mu$M) produced an increase in $[Ca]_i$ of 1730 $\pm$ 45%, similar to the elevation of $[Ca]_i$ seen when ionomycin 30 $\mu$M is added 30 min after a solvent blank (1390 $\pm$ 15%). This indicates that the reduced responses to AA and CA after desensitization were not due to non-specific effects of prolonged elevations of $[Ca]_i$ on the fluorescent dye or cells.

Previous studies have identified 3 intracellular, N-terminal cysteine residues required for hTRPA1 activation by CA (*Hinman et al., 2006*). We examined the effects of mutating all of these residues, Cys 621, Cys 641 and Cys 665 to serine on the activation of hTRPA1 by AA. As previously reported (*Hinman et al., 2006*), application of CA to the 3x Cys mutant hTRPA1 produced essentially no activation of the channel (Fig. 6). By contrast,

**Table 2 Activation of hTRPA1 by cinnamaldehyde or arachidonic acid inhibits subsequent agonist activation of the channel.** Changes in intracellular calcium concentration were determined as outlined in the Methods. Maximally effective concentrations of cinnamaldehyde (CA, 300 $\mu$M) or arachidonic acid (AA, 100 $\mu$M) were applied to HEK 293 cells expressing hTRPA1. Either CA or AA was then applied 30 min later. The first exposure to each agonist essentially abolished the subsequent response. The values represent the mean $\pm$ s.e.m. of the percent changes in raw fluorescence units, $n = 3$–5 determinations per condition.

| Drug additions | 1st addition (change in RFU, % predrug) | 2nd addition (change in RFU, % predrug) |
| --- | --- | --- |
| CA then AA | $506 \pm 31$ | $32 \pm 31$ |
| AA then Ca | $486 \pm 47$ | $42 \pm 14$ |
| CA then CA | $560 \pm 20$ | $-3 \pm 6$ |
| AA then AA | $486 \pm 43$ | $57 \pm 44$ |

the effects of AA were reduced but not abolished in the 3x Cys mutant hTRPA1 (Fig. 6). In these studies we extended the AA concentration response curve to include a concentration of 100 $\mu$M, the resulting $EC_{50}$ in wt hTRPA1 was $13 \pm 4$ $\mu$M, with a maximum increase in fluorescence of $390 \pm 70\%$. AA (100 $\mu$M) increased fluorescence in 3x Cys mutant hTRPA1 by $155 \pm 30\%$. NPPB is another agonist of TRPA1 which does not bind to the reactive cysteine residues (*Liu et al., 2010*), NPPB-induced elevations of calcium were also reduced but not abolished in the 3x cysteine mutant of hTRPA1 (Fig. 6). These data suggest that mutation of the cysteine residues can reduce but not abolish the activation of TRPA1 by unreactive compounds not structurally related to AA.

## DISCUSSION

The principle finding of our study is that AA and NAANs activate human TRPA1, although NAAN do so less effectively than AA. The activation of TRPA1 by AA and related compounds has a distinct profile from that reported for these compounds at other ion channels such as TRPV1 or Ca$_V$3 calcium channels. We also found that the AA activation of TRPA1 is only partially dependent on the presence of 3 N-terminal intracellular Cys residues that are required for activation of the channel by reactive electrophiles such as cinnamaldehyde. Our data is broadly consistent with studies reporting that AA (10 $\mu$M, (*Bandell et al., 2004*)) and docosohexanoic acid (*Motter & Ahern, 2012*) activate rodent TRPA1, but there appear to be differences in the effects of fatty acids and related compounds at human TRPA1.

We are confident that the effects of AA were being mediated by direct activation of TRPA1, and not by AA metabolites or though unspecific actions of AA on [Ca]$_i$. Under our experimental conditions, AA produced negligible increases in [Ca]$_i$ in untransfected HEK 293 cells, or in HEK 293 cells where TRPA1 expression had not been induced by tetracycline. Further, the effects of AA were blocked by specific (HC-030031, *McNamara et al., 2007*) and non-specific (ruthenium red) TRPA1 antagonists. AA is the parent molecule of a number of reactive compounds that can activate TRPA1 (e.g. *Materazzi et al., 2008*; *Taylor-Clark et al., 2008*; *Gregus et al., 2012*; *Sisignano et al., 2012*), however, the effects of AA were not modified by preincubation of HEK 293 cells with inhibitors of

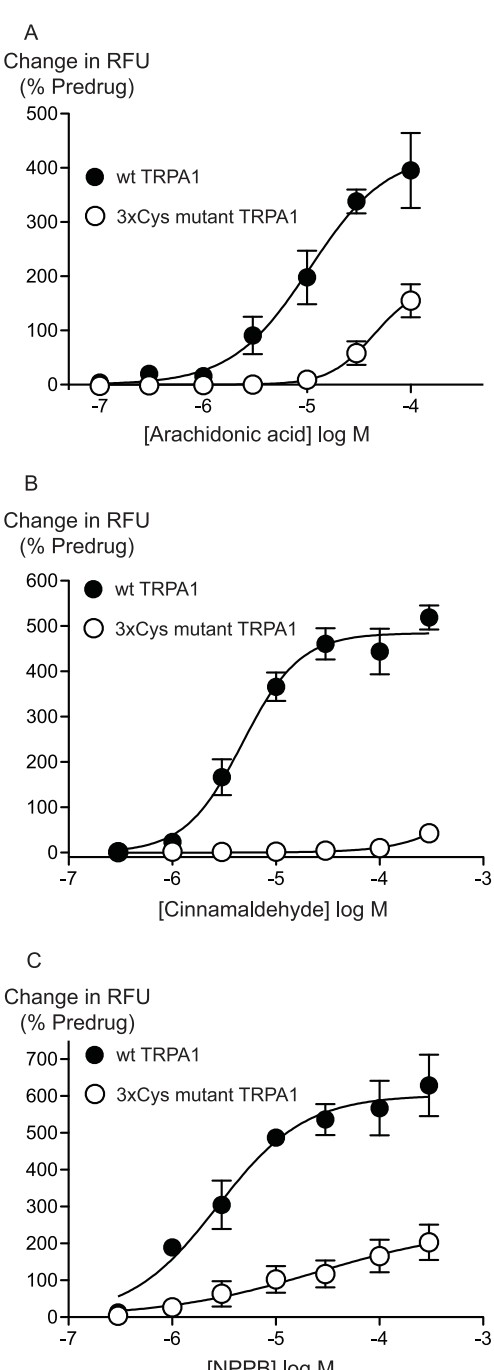

**Figure 6 Distinct amino acid residues in hTRPA1 determine arachidonic acid and cinnamaldehyde activation.** Changes in intracellular calcium concentration were determined as outlined in the Methods. Mutation of Cys621, Cys641 and Cys665 to Ser (3xCys mutant) prevents (A) cinnamaldehyde activation of hTRPA1 but only abrogates (B) arachidonic acid and (C) NPPB activation of the channel. Each point represents the mean $\pm$ s.e.m. of at least 5 independent determinations in triplicate. RFU = raw fluorescence units.

cyclo-oxygenase, lipoxygenase and FAAH, making it likely that AA itself was activating TRPA1. It is worth noting that the potency of AA to activate hTRPA1 is very similar to those of prostaglandin-derived TRPA1 agonists identified in previous studies (*Taylor-Clark et al., 2008*; *Materazzi et al., 2008*), but it is significantly less potent than hepoxilin A3 (*Gregus et al., 2012*) or 5,6 epoxyeicosatrienoic acid (*Sisignano et al., 2012*). Thus, metabolism of AA by 12-lipoxygenase or cytochrome P450 epoxygenase may increase the tissue availability of TRPA1 activators, while metabolism via cyclooxygenase is unlikely to, unless the derivatives were substantially more stable than AA, or if they were selectively available in a tissue compartment where AA levels were low.

Arachidonic acid was the most potent activator of hTRPA1 of the fatty acids we examined. In a study published while the present work was in preparation, (*Motter & Ahern, 2012*) used electrophysiological techniques to examine the effects of fatty acids on TRPA1. They focussed on docosahexaenoic acid as their reference compound. DHA activated rTRPA1 with an $EC_{50}$ between 11 and 40 $\mu$M, depending on the membrane potential where activation was measured. At a concentration of 100 $\mu$M, AA produced a similar increase in current to DHA, and this increase was similar to that produced by a high concentration of AITC (1 mM), a covalent TRPA1 agonist. In our experiments, which measured elevations in $[Ca]_i$ produced by activation of hTRPA1, the $EC_{50}$ of AA was about 10 $\mu$M. Interestingly, at 100 $\mu$M, DHA produced significantly less activation of both human and mouse TRPA1 than AA, while adrenic acid (C22:4) was inactive. Fatty acids with shorter acyl chains were also much less active than AA. We also found that $\omega$3-AA was much less effective than $\omega$6-AA at both human and mouse TRPA1. Interestingly, the carboxylic acid moiety of AA appears to be unnecessary for activation of TRPA1, as AA-ME was almost as equally effective as AA, and arachidonoyl ethanolamide and other NAAN retained substantial TRPA1 agonist activity.

*Motter & Ahern (2012)* did not directly compare the potencies of different fatty acids at TRPA1, and our results are largely consistent with theirs, with the exception of the relatively low activity of $\omega$3AA in the present study. It should be emphasized that there are significant differences in the methodology between the two studies. Firstly, our population measurements of TRPA1 activation were conducted at physiological temperature (37°C), a temperatures close to that at which TRPA1 undergoes temperature-dependent inactivation (*Wang et al., 2012*), while the study of Motter and Ahern was done at room temperature, conditions which may favour ligand activation of the channel. Secondly, our work measures both calcium influx through TRPA1 and any subsequent release of intracellular calcium or calcium entry from outside the cell produced by this, which may have an amplifying effect on the signal. Thirdly, our studies were done with the membrane potential of the cells free to vary between the resting potential of CHO cells and the reversal potential of TRPA1 (around 0 mV), the study of (*Motter & Ahern, 2012*) showed some voltage-dependence in the potency of DHA, with the compound being more potent at highly depolarized potentials. Finally, our work is done in intact cells, which may allow distinct mechanisms of channel modulation to those happening in cells subject to whole cell patch-clamp recordings. Nevertheless,

despite the recognized differences in pharmacology between rodent and human TRPA1, arachidonic acid and related compounds appear to act in a qualitatively similar manner.

$N$-acyl amino acids are a large family of lipid mediators that affect a variety of ion channels and receptors important for nociception (*Connor, Vaughan & Vandenberg, 2010*). None of the NAAN tested in the present study were as effective as AA or AEA in activating TRPA1. The most effective was NA-Tyr, with NA-Gly and NA-Tau being less active and NA-DA, NO-DA and NA-5HT being essentially inactive. This profile is quite distinct from that of these compounds at other well characterized effectors, TRPV1 and $Ca_V3$ channels. Notably, NA-DA and NO-DA are agonists at TRPV1 (*De Petrocellis et al., 2004*), while NA-5HT and NA-Tyr are antagonists (*Maione et al., 2007*, Connor, M *et al.* unpublished observations). Neither NO-DA nor NA-5HT inhibited the effects of AA at hTRPA1, suggesting that they interact with TRPA1 very weakly if at all. When considering NAAN modulation of $Ca_V3$ channels, both NA-5HT and NA-DA inhibit these channels with sub-micromolar potencies, as does AEA (*Chemin et al., 2001*; *Ross, Gilmore & Connor, 2009*; *Gilmore et al., 2012*). AA also inhibits $Ca_V3$ channels, but less potently than AEA, NA-5HT or NA-DA (*Chemin, Nargeot & Lory, 2007*; *Ross, Gilmore & Connor, 2009*; *Gilmore et al., 2012*). The rank order of effectiveness for fatty acid inhibition of human $Ca_V3$ channels, C22:6 ≈ C22:4 ≈ C20:4 > C20:2 > C20:1 > C20:4-methyl ester; (*Chemin, Nargeot & Lory, 2007*)) is quite distinct from that for activation of TRPA1, where C20:4 ≥ C20:4-methyl ester > C22:6 ≫ C22:4 ≈ C20:2 ≈ C20:1. The binding site for AA and related compounds has not been identified on either TRPA1 or $Ca_V3$ channels, but the distinct ligand/activity profiles at these channels suggests specific sites of interaction rather than un-specific interactions with the lipid membrane. This idea is reinforced by the very limited effects of the membrane fluidity-modifying detergent Triton-X 100 on TRPA1 (*Motter & Ahern, 2012*).

Several regions of TRPA1 have been shown to interact with ligands. The N-terminal ankyrin repeat domain of hTRPA1 is of major importance for the binding of reactive compounds, with three specific cysteine residues, Cys 621, Cys 641 and Cys 665 identified as crucial for channel activation by AITC and CA (*Hinman et al., 2006*). Mutation of these cysteines and lysine 708 also prevented activation of TRPA1 by 4-hydroxynonenol (*Trevisani et al., 2007*). By contrast, menthol and thymol agonist activity is dependent on specific residues in the fifth transmembrane domain (TM5) of hTRPA1 (*Xiao et al., 2008*) while the channel domains required for hTRPA1 activation by NPPB and farnesyl thiosalicylic acid remain incompletely defined. AA most resembled NPPB in that it retained significant activity in hTRPA1 where Cys621, Cys641 and Cys 665 had been mutated to serine. This channel was largely insensitive to CA. The requirement for intact Cys621/641/665 for full agonist activity of AA and NPPB has not been reported before, but likely reflects the importance of Cys-Cys cross-links involving Cys621 and Cys665 and other N-terminal Cys residues in maintaining the conformation of TRPA1 (*Wang et al., 2012*), rather than indicating that AA or NPPB covalently modify hTRPA1. Our data are consistent with those of (*Motter & Ahern, 2012*), who showed that the presence

of the N-terminal domain of murine TRPA1 was necessary but not sufficient for activation of mouse/drosophila TRPA1 chimeras.

Arachidonic acid is major signalling molecule derived from the actions of phospholipase $A_2$ on membrane phospholipids, and it acts directly on a diverse range of ion channels as well as serving as a precursor for a host of other molecules which activate or inhibit ion channels. Based on the affinity of AA for cyclooxygenase and lipoxygenase enzymes it has been suggested that concentrations of AA up to about 30 $\mu$M may be physiologically relevant (*Attwell, Miller & Sarantis, 1993*). TRPA1 is strongly expressed in subpopulations of sensory neurons and various epithelial cells throughout the body (*Bodkin & Brain, 2011*). There is also evidence for TRPA1-mediated modulation of neurotransmission in brain (*Shigetomi et al., 2011*). Thus, AA actions at TRPA1 could potentially modulate peripheral nociception, central neurotransmission, as well as lung, bladder and cardiovascular function. Our data suggests that there is a specific site where long chain fatty acids or endocannabinoids can interact with and activate TRPA1. Whether this site is the same as that utilized by arachidonic-acid derived molecules is unknown, although it is tempting to speculate that there may be an agonist site utilized by AA that may also provide a binding pocket for AA-derivatives such as 5,6 EET to facilitate their access to the N-terminal Cys residues of TRPA1 required for their activity (*Sisignano et al., 2012*). The definition of the AA binding determinants of TRPA1 may provide insights not only into how this channel is activated, but also how novel antagonists may be developed.

**Abbreviations**

**AA**  arachidonic acid (20:4 $\omega$6)

**AEA**  arachidonoyl ethanolamide

**AITC**  allyl isothiocyanate

**CA**  cinnamaldehyde

**[Ca]$_i$**  intracellular calcium

**DHA**  docosohexaenoic acid

**HBS**  HEPES buffered saline

**hTRPA1**  human transient receptor potential cation channel subfamily A, member 1

**NAAN**  $N$-acyl neurotransmitter/amino acid conjugate

**NA-5HT**  $N$-arachidonoyl serotonin

**NA-DA**  $N$-arachidonoyl dopamine

**NA-Gly**  $N$-arachidonoyl glycine

**NA-Tyr** *N*-arachidonoyl tyrosine

**NA-Tau** *N*-arachidonoyl taurine

**NPPB** 5-Nitro-2-(3phenylpropylamino)benzoic acid

### Funding

This work was supported by NHMRC Project Grant 1002680 to MC Sr and PM. WR and MC Jr were supported by Macquarie University International Research Scholarships, and received top up funding from the Australian School of Advanced Medicine. The funders had no role in study design, data collection and analysis, decision to publish, or preparation of the manuscript.

### Grant Disclosures

The following grant information was disclosed by the authors:
NHMRC Project Grant 1002680.

### Competing Interests

The authors declare that they have no competing interests.

### Author Contributions

- William John Redmond performed the experiments, analyzed the data, wrote the paper.
- Liuqiong Gu and Maxime Camo performed the experiments, analyzed the data.
- Peter McIntyre conceived and designed the experiments.
- Mark Connor conceived and designed the experiments, wrote the paper.

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
