# Peer review of "Ligand determinants of fatty acid activation of the pronociceptive ion channel TRPA1"

_PeerJ, doi:10.7717/peerj.248_

## Round 0.1 · original submission · Minor Revisions

· Academic Editor

Minor Revisions

Whilst the reviewers have made a number of detailed points regarding the submitted manuscript, all seem pertinent and justified.

Reviewer 1 ·

Basic reporting

- Line 71, please cite Karashima et al. (J Neurosci., 2007) for the effect of menthol on TRPA1.
- Lines 74-76. The statement on the reduction of the responses to “unreactive” compounds by cysteine mutations is not supported at all. This should be properly backed up with a thorough review of the literature before stating that the cysteine residues have “an important general role”.

Experimental design

- Line 133, I believe that the use of data obtained with cinnamaldehyde to complete the concentration-response curves of compounds of limited solubility is not appropriate. This “trick” assumes that these compounds and cinnamaldehyde have the same efficacy, but this does not have to be the case.
- Line 208, please note that 300 µM cinnamaldehyde does not induce maximal activation of mTRPA1 (Alpizar et al., Pflugers Archiv, 2013). The normalization of the effects of fatty acids to those of cinnamaldehyde assumes that the latter has the same effect on mouse and human isoforms. Is that the case?

Validity of the findings

- Figure 2, please show error bars in panel C.
- Line 180, please show a representative experiment for the block by RR.
- Figure 4, please show representative examples. Otherwise the bar plot alone does not say much and seems to be a waste of space.

Additional comments

- Line 108, delete the period after “mosmol)”

·

Basic reporting

No issues

Experimental design

No issues

Validity of the findings

The major conclusion that AA…“acts via a mechanism distinct from that of cinnamaldehyde, further underscoring the likelyhood (sic, should be likelihood) of multiple pharmacologically exploitable sites on hTRPA1” is unproven. More correctly, the data shows that the cys mutant has reduced responses to AA and, perhaps more dramatically to, CA, but nothing more. This applies equally to Discussion ln 263 where there is no evidence that the cys residues are “absolutely” required. The shifts in the concentration-relationship can be discussed, but there is no real evidence for differentially mechanisms.

Additional comments

This manuscript is a thorough investigation of the fatty acid TRPA1 pharmacology, with a useful comparison of arachidonic acid (AA), other fatty acids and N-arachidonoyl neurotransmitter/amino acid conjugates (NAAN) using calcium imaging and patch clamp electrophysiology. The study also investigates effects of previously reported mutations on fatty acid function. The study is of importance, although the physiological relevance could be better conveyed, and well performed throughout. As such I have only a few suggested points that would improve the manuscript to a general audience.

Major

1. The Abstract suffers in places from not reflecting the manuscript fully. Firstly, the statement that the cys-mutant cannot be activated by 300 microM cinnamaldehyde (CA) is not true, there is an (albeit small) effect. Also here, the first sentence of the Discussion identifies studies on NAAN function as a major finding of this study, the Abstract should better reflect this i.e define NAAN in the Abstract and state effects.

2. Whilst the Results do not need to veer too much into ‘Discussion’, it is useful to give the meaning of some data not discussed later. For example, what do the experiments described in ln 172-174 and ln 255-257 tell us?

3. The authors use ruthenium red (RR) as a non-selective and HC-030031 (HC) as a selective TRPA1 antagonist, respectively. It does appear that HC has a competitive effect, the authors should comment on the general TRPA1 antagonist pharmacology exhibited in this study as this is an area that requires development in the future. Related to this it would be useful to test antagonist effects in human TRPA1 and/or at least some of the extensive agonists used here.

4. The lack of effect of enzyme inhibitors on AA-induced changes to intracellular calcium levels have no positive controls, how can the authors be sure that the inhibitors are functional?

5. The experiments with the mutant channel are interesting, however, the justification for choosing the specific triple mutant should be given (preferably in the Results). For example, what were the characteristics of the triple mutant in terms of basal expression/conductance etc, why were these cys residues chosen, why were cys-ser mutations used and not cys-ala. Related to this, ln 249, add the reference for the “previously reported” statement.

6. The comparison with a similar paper from Motter and Ahern is justified but I found some points questionable (ln 300 onwards) and this section should be shortened/better justified. In particular, the second point applies equally to the Motter and Ahern study and should be deleted/revised and the final point is also unclear – the Motter and Ahern also predominantly use whole cell recording.

7. An important point is one regarding the physiological relevance of the present study in terms of endogenous concentrations of AA and TRPA1 activation. The authors allude to this themselves in ln 357-358, which is somewhat a negative way to finish the Discussion. What are the endogenous levels and could accumulation of fatty acids fit with reported EC50 values here? Finally, some idea about regions of TRPA1 expression would help promote physiological relevance of these data.

Minor
Introduction: ln 81 Motter & Ahern, 2012, also citation varies between ‘Motter & Ahern, 2012’ and ‘Motter and Ahern, 2012’ throughout.

Methods: state holding potential for electrophysiology experiments in Methods
ln 126-127: “All solutions had final ethanol concentration of 0.1% v/v” vs ln 141-142 “All drugs were made up in ethanol and diluted in HBS to give a final concentration of ethanol of 0.05 - 0.1%.”; which is correct?

Figure 2C: cinnamaldehyde should be labeled CA (not CIN) for consistency

---

## Round 0.2 · accepted · Accept

· Academic Editor

Accept

Thank you for attending to the comments raised by the reviewers in such a comprehensive fashion.

Reviewer 1 ·

Basic reporting

No further comments

Experimental design

No Comments

Validity of the findings

No Comments

Additional comments

No Comments

·

Basic reporting

No further comment

Experimental design

No further comment

Validity of the findings

No further comment